# Humoral Response and Safety after a Fourth Dose of the SARS-CoV-2 BNT162b2 Vaccine in Cancer Patients Undergoing Active Treatment—Results of a Prospective Observational Study

**DOI:** 10.3390/vaccines12010076

**Published:** 2024-01-12

**Authors:** Chiara Citterio, Claudia Biasini, Camilla Di Nunzio, Giuliana Lo Cascio, Luigi Cavanna

**Affiliations:** 1Oncology and Hematology Department, Piacenza General Hospital, Via Taverna 49, 29121 Piacenza, Italy; c.citterio@ausl.pc.it (C.C.); c.biasini@ausl.pc.it (C.B.); c.dinunzio@ausl.pc.it (C.D.N.); 2Clinical Pathology Department, Piacenza General Hospital, Via Taverna 49, 29121 Piacenza, Italy; g.locascio@ausl.pc.it; 3Casa di Cura Piacenza, Internal Medicine and Oncology, Via Morigi 41, 29121 Piacenza, Italy

**Keywords:** COVID-19 vaccines, safety, efficacy, cancer, booster

## Abstract

Only a few studies have been carried out on the efficacy and safety of a fourth dose of the COVID-19 vaccine in patients with cancer. In this prospective observational study, we aimed to assess the serological response and safety of the fourth booster shot of the BNT162b2 vaccine in 79 cancer patients, vaccinated between 1 March and 25 August 2022, under systemic anticancer therapy. The primary endpoint was to assess the increase in the anti-SARS-CoV-2 antibodies; secondary endpoints were the vaccine safety and side effects. Consequently, 40 patients (50.63%) revealed the maximum detection values in their IgG titers before the fourth dose of the vaccine, while 39 patients (49.37%) did not. Primary endpoint: Of 39 patients, 36 (92.31%) showed a significant increase in the anti-SARS-CoV-2 IgG titers, and 32 of them (82.05%) reached the maximum titration values. Secondary endpoints: The most common adverse events were mild in severity and included injection site pain, erythema and tiredness. The majority of the adverse reactions reported were grade 1 and no grade 3 and 4 reactions were detected. Our data provide evidence that a fourth dose of the BNT162b2 anti-SARS-CoV-2 vaccine is effective and safe in patients with solid tumors in active anticancer treatment.

## 1. Introduction

A novel coronavirus named Severe Acute Respiratory Syndrome Coronavirus 2 (SARS-CoV-2) emerged in Wuhan, China, in December 2019 and rapidly spread worldwide [1,2,3]. Patients with cancer are generally at high risk of severe infectious complications and mortality due to a poor general condition, the immunosuppression resulting from tumors and anticancer therapies such as chemotherapy, radiotherapy, surgery, steroids, etc. Moreover, hospital admission and many routine visits are a frequent occurrence among patients with cancer, with an increased risk of exposure to COVID-19 [4,5]. According to the Italian Ministry of Health and the Italian Drug Agency (AIFA), medically vulnerable groups, including cancer patients, should be prioritised for an mRNA-based vaccine, i.e., Pfizer/BionTech or Moderna [6]. In two of our previous prospective observational studies, we reported the seroresponse rate and safety after a two-dose regime of the BNT162b2 or mRNA-1273 vaccine in adult patients with solid cancer undergoing active anticancer treatment: both the BNT162b2 and mRNA-1273 vaccines provided highly efficient protection and showed a very good safety profile in patients with cancer [7,8]. Our first observational prospective study showed that after two doses of the BNT162b2 or mRNA-1273 vaccine, the antibody response rate in 219 cancer patients undergoing active anticancer treatment was detected to be 75.88%. The vaccination was well tolerated in this group and no grade 3 and 4 complications were detected [7]. As older patients with cancer are at high risk of COVID-19, a second observational prospective study was conducted to evaluate the seroconversion rate and safety in response to two doses of the BNT162b2 or mRNA-1273 vaccine involving 115 patients aged >70 years [8]. This group of patients included 64 females (55.65%), 21 patients (18.26%) with haematological malignancies and 94 patients (81.74%) with solid tumors. This study assessed that the seroconversion rate was detected to be 65.22% in patients with solid tumors and 42.86% in patients with haematological malignancies. The COVID-19 BNT162b2 and mRNA-1273 vaccines have been found to be safe and effective in patients with cancer aged >70 years and undergoing active anticancer treatment [8]. Other studies have shown that a third dose of COVID-19 vaccines strongly boosts the waning immune system in cancer patients [9,10]. A systemic review has demonstrated that booster doses of COVID-19 vaccines are effective in improving seroconversion and antibody levels; however, patients with haematological cancer consistently demonstrated a poorer response to booster vaccines than patients with solid cancers [9]. Similar results after an early COVID-19 booster dose were reported by Samol J. et al. [10]. However, there is a paucity of data on the serological response to and safety of a fourth dose of an mRNA COVID-19 vaccine in patients with cancer in active treatment [11,12]. This observational prospective study addresses the antibody-mediated response and safety of a BNT162b2 (Pfizer/Biontech) fourth vaccine dose in cancer patients undergoing active anticancer treatment. We believe that our article differs from previous studies, as it is a prospective observational study, and in addition, it describes the response to the COVID-19 mRNA vaccine in patients with cancer treated with chemotherapy, biological therapy, immunotherapy and hormone therapy, in different settings of the disease, from the neoadjuvant to metastatic phase, as is currently applicable in the real world of oncology practice.

## 2. Materials and Methods

In this study, conducted in the Oncology–Hematology Department of Piacenza General Hospital (Northern Italy) and approved by the local ethics committee (Institutional Review Board approval number 2022/0269473), we investigated the safety and immunogenicity of the fourth dose of a SARS-CoV-2 mRNA-based vaccine in patients with solid tumors undergoing active treatment or who had ended cancer therapy within 6 months before this study period and with no known history of SARS-CoV-2 infection. Eligible individuals had to receive a third dose regime of the BNT162b2 mRNA vaccine (Pfizer/Biontech) or the mRNA-1273 vaccine (Moderna) at least 120 days before the fourth dose. The vaccine was injected into the deltoid muscle in accordance with the manufacturer’s technical instructions and was administered to those patients undergoing cytotoxic chemotherapy, biological therapy (such as monoclonal antibodies trastuzumab, bevacizumab etc.), tyrosine kinase inhibitors, hormone therapy or immune checkpoint inhibitors, as recommended [13]. Vaccination uptake was provided by the Public Health Services in Piacenza in accordance with patients’ informed consent. Before vaccination, a complete blood count was conducted, and it was delayed if necessary until neutrophil count recovery. The blood serum of the cancer patients was tested to evaluate the serum IgG antibody levels against SARS-CoV-2 up to two days before vaccination (TO) and 2–8 weeks after the fourth vaccine dose (TI). We did not analyse the expression of cytokines in the patient’s serum before and after the fourth dose.

### 2.1. Endpoints

The primary endpoint was the increase in the anti-SARS-CoV-2 S antibodies (in patients with a value inferior to the highest detection level); the secondary endpoints included the safety and side effects of the vaccine. In order to record accurately the vaccines’ side effects, all the patients were informed to register and to call their oncologist or specialised nurses to report any adverse events that followed vaccination.

### 2.2. Serologic Assessment

The serum samples were collected and analysed for detecting SARS-CoV-2 antibodies against spike (S) protein S1 and S2 antigens using the serologic automated assay LIAISON XL SARS-CoV-2 S1-S2 IgG [14,15]. The results were expressed in binding antibody units (BAU/mL), referring in comparison to a calibration curve. According to the manufacturer’s instructions, the maximum detection limit goals were set at 400 BAU/mL, and a concentration of >15 BAU/mL was considered positive [14,15]. The participants were divided into two groups; group A included patients with the maximum detection limit of IgG prior to the fourth vaccine dose, while group B included patients without the maximum detection limit of IgG prior to the fourth dose.

### 2.3. Statistical Analysis 

In the patient registration process, each patient was assigned a unique QR code identity tag in a Microsoft Excel file (Microsoft Office version 2010). Median and interquartile range (IQR) were used to calculate the quantitative data. Absolute and percentage frequencies described the qualitative data. The normal distribution was checked for all continuous variables. All statistical analyses were performed using the RStudio (version 3.6.0) statistical software with a Sig. (2-tailed test) value with a significance level set to 5%.

## 3. Results

A total of 79 cancer patients received the fourth dose of the BNT162b2 vaccine between 1 March 2022 and 25 August 2022. All of them had received a third dose of the COVID-19 vaccine at least four months (range 4–6 months) before the fourth dose. The median age was 73 (IQR 65–78.5) years, with a range between 32 and 90 years in the overall population, and 50.63% were female (Table 1). The majority of patients showed metastatic disease (70.89%), while the most common cancers were gastrointestinal (35.44%), genitourinary (21.52%) and breast (15.19%). Other cancers included pancreas, lung and head–neck (Table 1). The majority of the patients received chemotherapy (53.16%). Equally, 40 (50.63%) patients showed the maximum detection values (400 AU/mL of IgG titers) prior to the fourth dose of the vaccine (Group A), while 39 (49.37%) patients showed IgG levels < 400 AU/ML (median level of IgG 172.18 AU/mL) (Group B). Primary endpoints: After the fourth dose of the COVID-19 vaccine, 32 out of 39 patients (82.05%) who did not reach the maximum detection values (400 AU/mL) prior to the fourth dose of the vaccine revealed a significant increase in their anti-SARS-CoV-2 IgG titers and reached the maximum titration values. Furthermore, 4 out of these 39 patients (10.26%) showed an increase in IgG titers (median increase of 134.25 AU/mL) but did not reach the maximum level of 400 AU/mL, and 3 patients (7.69%) did not show any increase in IgG titers, while the range of IgG values in patients’ responses was 18.5–375 AU/mL (Table 2). Secondary endpoints: The most common adverse events were mild. Injection site pain was reported in 35 patients (44.30%) and local erythema in 43 patients (54.43%). Overall, 40 patients (50.63%) reported at least one systemic adverse event, and all were mild: tiredness in 23 (29.11%), headache in 10 (12.66%) and a runny nose in 6 (7.60%). Only three patients (3.80%) reported fever and four (5.06%) myalgia. The majority of the side effects were of grade 1, while no grade 3 or 4 side effects were recorded (Table 3). 

## 4. Discussion

Patients with cancer are generally at an increased risk of developing infectious complications. Given that, COVID-19 containment measures are recommended, and are being implemented to prevent the infections both of patients and healthcare professionals. The COVID-19 pandemic exploded first in China, and subsequently in Europe. Italy was the first European country to be affected severely and to attain very high incidence rates. By 1 April 2020, within a month from the first case reported in Lombardy (Northern Italy), 100,000 cases and more than 12,000 deaths had been registered in the country [16]. Out of 909 patients who died in hospital, 150 patients had active cancer over the past 5 years [16]. The Italian government, following the Chinese model, established contingency response measures to reduce the risk of COVID-19 spreading. As cases across the country hit a new daily record, the first National Decree was issued on 8 March 2020, and three Italian regions (Emilia Romagna, Lombardy and Veneto) were soon declared ‘red zones’ and put under new restrictions. In this extraordinary worldwide situation, it is relevant to consider the strong impact that COVID-19 had on healthcare professionals, resources and staff, as well as massive consequences for cancer patients in terms of screening, diagnosis, treatment and oncology clinical trials. In Italy, there is active screening for breast cancer, colorectal cancer and cervical cancer. A national survey carried out in 2020 by the Italian National Centre for Screening Monitoring (ONS) revealed that participation in these three main cancer screening programmes decreased significantly during the pandemic, respectively, by 37.6%, 45.5% and 43.4% in 2020 compared with 2019. The estimated numbers of undiagnosed lesions were 3.324 for breast cancer, 1.299 for colorectal cancer, 7.474 for colorectal advanced adenomas and 2.782 for CIN2 or more severe cervical lesions.

Piacenza was the closest city to the first COVID-19 cluster in Northern Italy and, during the first three months of the pandemic, suffered a particularly heavy burden. We investigated the impact of COVID-19 on the subsequent management of cancer patients in terms of diagnosis, types of cancer treatment and oncology clinical trials during this pandemic period of 2020 compared with 2019 [17]. The number of patients diagnosed and treated, as well as those enrolled in clinical trials, and the type of treatment received (intravenous vs. oral) were collected and compared with those in 2019. We observed a statistically significant decrease in the number of new cancer diagnoses and in new clinical trial enrollment, as well as a change in drug administration, with less intravenous and more oral drug administration [17]. Before any explanation can be found, it is essential to define the problem accurately. During the first COVID-19 wave, Italian hospitals and healthcare facilities faced an unprecedented massive inflow in a very short period of time. Almost all frontline healthcare providers such as doctors and nurses were being placed under work pressure and were directly engaged in departments dedicated to or where patients diagnosed with COVID-19 were hospitalised. In our district of Piacenza, our medical system was literally on the verge of collapse, and hospital beds for patients affected with SARS-CoV-2 infection were almost fully occupied, so that the remaining patients had to be admitted to nearby hospitals and clinics. Consequently, patients with other diseases, including cancer patients, experienced obstacles in their access to healthcare services and were thus unable to receive surgical care since medical oncology continued through oncologic therapy. In addition, patients would visit the hospital less frequently to avoid being infected with SARS-CoV-2. The oncologists at Piacenza General Hospital developed a strategic intervention to treat early COVID-19 patients at home, thus avoiding access to hospital [18]. Despite all these efforts, many patients receiving cancer care had their cancer surgeries and treatments cancelled or delayed. Compared to those infected during the first wave, the patients in the second wave in Piacenza died less frequently and were hospitalised less frequently (approximately 15% bed occupancy versus 95% in the first wave). Despite COVID-19 representing a public health threat, cancer still remains one of the main causes of death, and postponing or modifying treatment schedules may lead to worse outcomes for cancer patients [7,17].

Before the introduction of COVID-19 vaccines, widespread diagnostic testing was crucial in facilitating the early diagnosis of COVID-19 and maintaining the appropriate therapy for cancer patients [19]. In an area of high SARS-CoV-2 infection, like our district in Italy [20], we reported a prevalence of COVID-19 infection in asymptomatic cancer patients. From April to June 2020, 260 consecutive asymptomatic (for COVID-19) cancer patients were tested for COVID-19. This group of patients was made up of 160 women and 100 men; 218 patients were in active treatment, 32 were in the diagnostic stage waiting for treatment and 10 received supporting care only. A total of 10 out of the 260 patients (3.85%) tested positive. Routine asymptomatic testing was a crucial component of effective targeted control strategies for COVID-19 to help prevent outbreaks among the vulnerable population and care workers such as doctors and nurses. Oncology societies and national authorities such as the American Society of Clinical Oncology (ASCO), the European Institute of Oncology (ESCO), the Italian Association of Medical Oncology (Associazione Italiana di Oncologia Medica, AIOM) and the Italian Board of Medical Oncology Hospital Directors (Collegio degli Oncologi Medici Ospedalieri, CIPOMO) have been quick to issue guidelines on cancer care and cancer patient management during the pandemic.

Considerable efforts to protect vulnerable patients with cancer from COVID-19 infection have also been made by political institutions, the oncology community and family caregivers. During the COVID-19 pandemic, telemedicine became very important to reduce the risk of cancer patients being exposed to the SARS-CoC-2 infection.

A systematic review and meta-synthesis of the qualitative literature on the use of telemedicine in cancer care during the COVID-19 pandemic has been reported from Iran [21]. In this research, 19 studies were included in the final meta-synthesis, which concerned 684 healthcare providers, 256 patients, 16 caregivers and 1 patient advocate. This meta-synthesis highlighted that telemedicine was more effective in managing cancer patients’ primary healthcare needs and follow-up visits; in addition it contributed to recognising the advantages and disadvantages to, prerequisites for and preferences of patients and providers based on their lived experiences. Telemedicine during the COVID-19 pandemic was rapidly incorporated at the oncology division of Tel Aviv Medical Center and a survey was undertaken among adult patients with cancer treated there between March and May 2020 [22]. The results of this survey showed that 232 patients used a telemedicine platform between March and May 2020, and 172 (74%) agreed to participate in the survey. Family members/caregivers were commonly present during telemedicine meetings. It is very important to make evident that in this study, a multivariate analysis revealed that higher satisfaction and visits for routine surveillance were both predictors of willingness to continue future telemedicine meetings over physical encounters and telemedicine was perceived as safe and effective. A study from France [23], based on a qualitative method with semidirected interviews with doctors from the oncology and supportive care departments of Foch Hospital, which had used telemedicine during the first wave of the COVID-19 pandemic, showed a large difference between doctors in terms of their views concerning telemedicine before and after the first wave of the COVID-19 pandemic in France. However, during the COVID-19 pandemic, many providers and patients had to adopt the use of telemedicine in a relatively short time, and it was hypothesised that socioeconomic differences could create gaps in telemedicine adoption among different patient groups during this extraordinary and dramatic period [24].

The COVID-19 pandemic is an unprecedented challenge with an immediate impact on cancer patients and created huge demand for supportive cancer care interventions for these patients, who experienced overwhelming stress during trying times and now need healthcare prioritisation. Policy should focus on the development of instruments for efficient cancer programmes, which would encompass the cancer care model, the operational requirements, as well as financial implications and both short- and long-term strategy considerations.

In 2021, the BNT162b2 and mRNA-1273 vaccines were approved by the United States Food and Drug Administration (FDA) and the European Medicine Agency, and first became available. However, given that cancer patients were not included in the pivotal clinical trials, considerable uncertainty remains regarding the vaccine efficacy, as well as the tolerability and safety of COVID-19 vaccines, in cancer patients. In phase III trials, both lipid nanoparticle-encapsulated mRNA-based vaccines reached, respectively, 94.00% and 95% efficacy in preventing symptomatic SARS-CoV-2 infection independently of age. The vaccination of cancer patients, included those in active treatment, was then recommended by major Western oncology societies such as the American Society of Clinical Oncology, the American Association for Cancer Research, the Association of American Cancer Institutes, the European Society of Medical Oncology, the Society for Immunotherapy of Cancer and the Italian Association of Medical Oncology [7]. 

It is well known that the prevention of the higher risk of severe COVID-19 and poor outcomes is crucial for patients with cancer, with vaccination being the most effective method for achieving this goal [9,11]. The clinical efficacy of COVID-19 vaccination in reducing the rates of symptomatic infections and severe outcomes has been described [9,10,11,12] and the results of third and fourth boosters have been reported to enhance protection in almost all immunocompromised patients [9]. However, there is a paucity of evidence that a fourth booster dose of COVID-19 vaccines is effective in improving antibody levels and is safe in patients with solid tumors and in active treatment. Mai AS et al. [9], in a systematic review and individual patient data meta-analysis, reported data on booster doses of COVID-19 vaccines in 849 patients with haematological cancer and in only 82 patients with solid cancer, and demonstrated that a COVID-19 vaccine booster dose is effective in improving seroconversion and antibody levels. It must be emphasised, however, that this systematic review and meta-analysis evaluated the results of a third booster, but not of a fourth dose. A recent systematic review of 24 published studies on 2838 patients that reviewed data on the administration of a fourth COVID-19 vaccination in immunocompromised patients [12] demonstrated an increase in antibody titers after receiving a fourth dose. Immunocompromised patients included solid organ transplant recipients, patients with autoimmune rheumatic disease, patients with human immunodeficiency virus (HIV) and patients with blood cancers or disease; almost all the patients reported in this review received the BNT162b2 and mRNA-1273 vaccines as a fourth dose, but patients with solid tumors submitting to a fourth vaccine dose are lacking from this review. The immunological response after the fourth BNT162b2 COVID-19 vaccine has been reported in a limited series of 23 patients with solid tumors, with a median age of 66 years. These patients were in active treatment, the majority of them had lung cancer (82%) and they were treated with immunotherapy (91%). All but one of these patients, three weeks after the fourth dose, developed positive levels of anti-spike antibodies [25]. Recently, a nationwide cohort study from Singapore has provided evidence of the clinical effectiveness of mRNA-based fourth booster dose vaccines against COVID-19 in patients with cancer [26]. However, this study revealed some limitations, as reported by the authors themselves: the study may have been objected to confounding variables that could not be controlled for [26]. In our hospital, a task force charged with promoting the importance of the COVID-19 vaccine for cancer patients has been established since February 2021. Local communities of patients with cancer, local television, as well as radio campaigns and healthcare professionals were also involved in providing information on the benefits and risks of COVID-19 vaccines [27]. Following these strategies, only 3 (0.68%) out of 443 cancer patient candidates for vaccination refused the vaccine. In an observational prospective study, we reported the results of two doses of m-RNA COVID-19 vaccines in 257 evaluable cancer patients older than 70 years [8] undergoing active treatment.

The present study reveals that 92.31% of cancer patients in active treatment improved in their IgG values after a fourth dose of an mRNA-based vaccine; 82.05% of them reached the maximum values in their IgG titers. We are well aware that the correlation between the antibody response to an anti-COVID-19 vaccine and protection against COVID-19 infection has not been clearly established in patients with cancer. However, as previously reported [7,28], in cancer patients vaccinated against COVID-19, a significant reduction in clinically relevant infection has been observed.

## 5. Conclusions

We conducted a prospective observational study on 79 consecutive patients with cancer, all of them in active anticancer treatment: 42 patients (53.16%) with chemotherapy, 21 patients (26.58%) with biological therapy, 16 patients (20.25%) treated with other drugs. All 79 patients received a fourth booster dose of the BNT162b2 vaccine between 1 March 2022 and 25 August 2022. Interestingly, all the patients accepted the fourth dose of the BNT162b2 vaccine proposed by our healthcare professionals. This study showed data on the efficacy of a fourth booster of the BNT162b2 vaccine: 36 out of 39 patients (92.31%) who did not reach the maximum detection values (400 AU/mL) prior to the fourth dose of the vaccine exhibited a significant increase in anti-SARS-CoV-2 IgG titers and 32 patients (82.05%) reached the maximum titration value, while 4 out of 39 patients (10.26%) showed an increase in their IgG titers (median increase of 134.25 AU/mL) but did not reach the maximum level of 400 AU/Ml. Only three patients (7.69%) did not show any increase in IgG titers. In addition, our study reported data on the safety of a fourth BNT162b2 vaccine dose in patients with solid tumors in active anticancer treatment. The vaccine was well tolerated, and the adverse events were mild, such as local erythema in 43 patients (54.43%) and injection site pain in 35 patients (44.30%). Systemic adverse events were also mild, such as tiredness in 23 patients (29.11%), headache in 10 (12.66%) and a runny nose in 6 (7.06%). The majority of side effects were of grade 1. No grade 3 and 4 side effects were recorded. The side effects after the third and fourth dose of COVID-19 vaccine were mild and similar in our series of patients undergoing active treatment.

Our study also reveals the longevity of the SARS-CoV-2 antibody response in patients with cancer in active treatment starting from at least four months after the third dose. However, for vaccinated patients with cancer, especially those undergoing active cancer treatment, our recommendations on the use of facemasks when required, social distancing and hand hygiene practice are still emphasised. This study also reports data on the efficacy and safety of a fourth COVID-19 vaccine dose in patients with cancer and in active treatment with cytotoxic chemotherapy, immunotherapy and target therapy.

## Figures and Tables

**Table 1 vaccines-12-00076-t001:** Clinical and demographic characteristics.

Variable	Patients (n = 79)
Median age at time of vaccination [IQR] (range)	73 [65–78.5] (32–90)
Sex n (%)	
Female	40 (50.63)
Male	39 (49.37)
Primary tumor location n (%)	
Gastrointestinal	28 (35.44)
Genito-urinary	17 (21.52)
Breast	12 (15.19)
Pancreas	7 (8.86)
Lung	5 (6.33)
Head–neck	3 (3.80)
Other	7 (8.86)
Stage n (%)	
I–III	21 (26.58)
IV	56 (70.89)
NA	2 (2.53)
Therapy n (%)	
Chemotherapy	42 (53.16)
Biological therapy	21 (26.58)
Chemotherapy + biological therapy	2 (2.53)
Chemotherapy + Immunotherapy	2 (2.53)
Immunotherapy	3 (3.80)
Hormone therapy	9 (11.39)
Line n (%)	
Neoadjuvat/adjuvant	16 (20.25)
I line (Metastatic)	37 (46.84)
>I line	26 (32.91)

**Table 2 vaccines-12-00076-t002:** Fourth dose of COVID-19 vaccine’s effectiveness in cancer patients.

	N	%
Total vaccinated patients	79	100
Patients with maximum detection value IgG titers prior the fourth dose of vaccine (Group A)	40	50.63
Patients without maximum detection value IgG titers prior the fourth dose of vaccine (Group B)	39	49.37
− Median IgG value (range) (Group B)	172.18 AU/mL (18.5–375)	
− N. patients reaching maximum detection values IgG titers after the fourth dose of vaccine	32	82.05
− N. patients with increased IgG titer not reaching maximum detection values	4	10.26
− Median IgG value increase (range)	134.25 AU/mL (98–174)	
− N. patients with stable value IgG titers after the fourth dose of vaccine	3	7.69

**Table 3 vaccines-12-00076-t003:** Side effects of fourth dose of COVID-19 vaccine in cancer patients.

Symptom	Number (%)	Grade 1 (%)	Grade 2(%)	Grade 3 (%)	Grade 4(%)
local pain	35 (44.30)	80	20	0	0
local erythema	43 (54.43)	90	10	0	0
weakness	23 (29.11)	60	40	0	0
headache	10 (12.66)	80	20	0	0
runny nose	6 (7.60)	100	0	0	0
myalgia	4 (5.00)	90	10	0	0
fever	3 (3.80)	80	20	0	0

## Data Availability

The data that support the findings of this study are available on request from the corresponding author [L.C.].

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
