# Peer review of "Humoral Response and Safety after a Fourth Dose of the SARS-CoV-2 BNT162b2 Vaccine in Cancer Patients Undergoing Active Treatment—Results of a Prospective Observational Study"

_vaccines, 2024, doi:10.3390/vaccines12010076_

Round 1

Reviewer 1 Report

Comments and Suggestions for Authors

This study investigated the safety and efficacy of the fourth Covid-19 booster shot in cancer patients. Authors of this study recruited 79 cancer patients who had received three doses of BNT162b2 vaccine. The patients were given a fourth dose of the vaccine, and their blood was collected before and after the fourth dose. The blood was used to measure anti-SARS-CoV-2 IgG titers. The authors also collected information about the patients' adverse responses to the fourth dose.

The study provides useful information about the safety and efficacy of the fourth dose of the BNT162b2 vaccine in cancer patients. It should be accepted for publication barring a few minor corrections.

Suggested corrections:
- line 29: "SARS-Co-V-2" -> "SARS-CoV-2"
- line 61: "m-RNA" -> "mRNA"
- line 70: "m-RNA" -> "mRNA"
- line 81: "a complete blood count cell" -> "a complete blood count"

Author Response

We performed the corrections as suggested by the Reviewer: line 29: "SARS-Co-V-2" -> "SARS-CoV-2", line 61: "m-RNA" -> "mRNA", line 70: "m-RNA" -> "mRNA", line 81: "a complete blood count cell" -> "a complete blood count".

Reviewer 2 Report

Comments and Suggestions for Authors

The manuscript entitled: “Humoral response and safety after a fourth dose of 2 SARS-CoV-2 BNT162b2 vaccine in cancer patients undergoing 3 active treatment. Results of a prospective observational study” deals with a very interesting and important topic such as the efficacy and safety of 4th dose of COVID vaccines in patients with cancer.

It is a well written manuscript and the following minor changes will improve the final manuscript:

·      In title authors probably wanted to write “treatments” instead of “treatment”

·      Authors should pay attention to double spaces or no spaces in their manuscript.

Results section:

·      Apart from dividing patients into those presenting maximum or not detection limit of IgG, presenting the actual IgG values in BAU/ml would be more informative regarding the range of patients’ response.

Discussion section:

·      Discussion is quite long with many not necessary details. Authors should rewrite a shorter version omitting details regarding eg the effect of telemedicine in cancer patients’ healthcare that should be presented in am more concise way.

Comments on the Quality of English Language

English is generally of high level, minor editing is required.

Author Response

We performed the changes as suggested:

Title we write “treatments” instead of “treatment”, we payed attention to the spaces in our manuscript. Results section: the range of patients’ response was added at page 3, line 126: “the range of IgG values of patients’ response were 18.5-375 AU/ml”, before the word “Table 2”.

Discussion section: the paragraph of telemedicine was streamlined as suggested and the sentence “Health-care providers included leaders of cancer centres, home care professionals, social workers, nurses, physicians, care managers, pharmacists, psychologists, and other professionals in the oncology field” (lines 213-215 page 6) was cancelled.

Reviewer 3 Report

Comments and Suggestions for Authors

Known in the field based on previous literatures:

  1. Covid-19 is an infectious disease caused by severe acute respiratory syndrome coronavirus 2 (SARS-CoV-2). The disease had blowout worldwide in 2019 and symptoms of COVID-19 are variable, but often include cough, breathing difficulties, headache, fever, fatigue, loss of smell and taste.
  2. COVID-19 vaccine is developed and intended to provide acquired immunity against the virus.

In this observational article authors reported following findings:

I have gone through the article titled ‘Humoral response and safety after a fourth dose of SARS-CoV-2 BNT162b2 vaccine in cancer patients undergoing active treatment. Results of a prospective observational study’. Authors investigated the risk of adverse effects and humoral responses following immunization with mRNA COVID-19 vaccine after 4th dose of vaccine among cancer patients undergoing active treatment. Following are the main points-

  1. Covid-19 vaccine is effective, but several side effects were speculated and studied. In this article, authors studied the side effect of booster dose of COVID-19 vaccine and reported the local pain, erythema, weakness and headache after 4th immunization.
  2. Further, authors performed serological assessment of serum samples collected from patients and analyzed SARS-Cov-2 antibodies against spike proteins (S1 and S2).

The many facts and material presented are already available and there is nothing new except results of a prospective observation. Although, the data presented are interesting and supportive of the conclusions drawn. The following suggestions if incorporated could help in the better understanding of the significance of the work and implications.

Minor/Major Concerns:

1. What are the novel outcomes in this study except prospective observation? Authors should clearly mention about how this article different from rest? Does it embrace a specific gap in the field?

2.  Method is not sufficiently described. Please elaborate the described methods used in this study.

3. What was the similarity and dissimilarity of side effects after 3rd and 4th dose of Covid-19 vaccine in this study? Please discuss it or add a table.

4.  Have you analyzed the expression of pro or anti-inflammatory cytokines in the patient’s serum before and after 4th dose?

Author Response

  1. We added in the introduction section, page 2 line 64: “We want to make evident that our study represents the response to COVID-19 mRNA vaccine in patients with cancer treated with chemotherapy, biological therapy, immunotherapy, hormone therapy, in different settings of the disease, from neoadiuvant to metastatic phase as is currently the case in the real world of oncology practice.”.
  2. We added at page 3 line 100 “The patients were informed to call oncologists or specialized nurses to report any adverse event come later the vaccination”.
  3. We added at page 8 line 319 after the word “recorded”: “The side effects after 3rd and 4th dose of COVID-19 vaccine were mild and similar in our series”.
  4. “We did not analyzed the expression of cytokines in the patient’s serum before and after 4th dose” was added at page 2 line 84 after the words “vaccine dose (T1)”.